# Understanding the Medical Chemistry of the Cannabis Plant is Critical to Guiding Real World Clinical Evidence

**DOI:** 10.3390/molecules25184042

**Published:** 2020-09-04

**Authors:** Karim S. Ladha, Prabjit Ajrawat, Yi Yang, Hance Clarke

**Affiliations:** 1Department of Anesthesia and Pain Medicine, University of Toronto, Toronto, ON M5G 1E2, Canada; karim.ladha@mail.utoronto.ca; 2Department of Anesthesia, St, Michael′s Hospital, Toronto, ON M5B 1W8, Canada; 3Centre For Cannabinoid Therapeutics, Toronto, ON M5G 2C4, Canada; prab.ajrawat@mail.utoronto.ca; 4Department of Anesthesia, Toronto General Hospital, Toronto, ON M5G 2C4, Canada; 5Centre for Molecular Design and Preformulations and Krembil Research Institute, University Health Network, Toronto, ON M5G 1L7, Canada; William.Yang@uhnresearch.ca; 6Department of Pharmaceutical Sciences, Leslie Dan Faculty of Pharmacy, University of Toronto, Toronto, ON M5S 3M2, Canada; 7Transitional Pain Service, Toronto General Hospital, Toronto, ON M5G 2C4, Canada

**Keywords:** cannabis, real world evidence, Δ*^9^*-tetrahydrocannabinol, cannabidiol

## Abstract

While cannabis has been consumed for thousands of years, the medical-legal landscape surrounding its use has dramatically evolved over the past decades. Patients are turning to cannabis as a therapeutic option for several medical conditions. Given the surge in interest over the past decades there exists a major gap in the literature with respect to understanding the products that are currently being consumed by patients. The current perspective highlights the lack of relevance within the current literature towards understanding the medical chemistry of the products being consumed. The cannabis industry must rigorously invest into understanding what people are consuming from a chemical composition standpoint. This will inform what compounds in addition to Δ*^9^*-tetrahydrocannabinol and cannabidiol may be producing physiologic/therapeutic effects from plant based extracts. Only through real-world evidence and a formalized, granular data collection process within which we know the chemical inputs for patients already using or beginning to use medical cannabis, we can come closer to the ability to provide targeted clinical decision making and design future appropriate randomized controlled trials.

## 1. Introduction

While the use of cannabis for medicinal purposes dates back thousands of years, there has been a recent surge in interest over the past two decades surrounding its potential therapeutic properties. This renewed awareness has been reflected in a growing number of governments creating legal frameworks that authorize the use of cannabis for medical purposes. In the United States, the majority of states have now created pathways for medical cannabis [1] and Canada has had legislation in place since 2001 for physicians to authorize its use [2]. Yet this trend is not limited to North America. The European Parliament recently passed a motion to incentivize member states to increase access and research funding for medical cannabis. As societal attitudes towards cannabis shift, we in medicine must also adapt in order to provide quality care for our patients.

Regardless of a clinician′s personal opinions and beliefs, an increasing number of patients will undoubtedly inquire about using cannabis for a variety of ailments. Healthcare providers have a duty to inform themselves of the evidence related to medical cannabis in order to have the ability to appropriately discuss the risks and benefits—a standard that should be applied to any therapy. At the same time, an important question to ask is—does the current literature provide useful guidance as it applies to the forms of cannabis available today? In this review we argue that the current evidence-base lacks relevance and new data generated from prospective observational studies, that is, real-world evidence, is a crucial first step towards informing future trials.

## 2. Current State of Affairs

The problem with the current state of knowledge is not related to quantity. Indeed, there is no shortage of studies exploring the use of cannabis as a therapeutic agent. A quick search of PubMed will reveal numerous systematic reviews that have been published over the past five years exploring the curative properties of the plant. In fact, it is now possible to do a systematic review of systematic reviews [3]. Whether it be for noncancer [4] or cancer pain [5], psychiatric disorders [6] or athletic performance [7], the conclusions across these reviews are similar—clinical trial data with cannabinoid products demonstrates mild to no benefit. Evidence-based medicine rubrics typically place systematic reviews on a pedestal as the highest form of evidence. Therefore, we have a situation where the majority of evidence from the “pinnacle of the pyramid” is equivocal. At the same time, another quick search using Google will reveal numerous websites extolling the curative properties of cannabis. It may be easy to simply dismiss these claims as being scientifically unsound and indeed many claims including ones based on rigorous trials are often sensationalized [8]. However, it is important to first explore why there might be a discrepancy between anecdotal claims and randomized controlled trials.

While a thorough analysis of the cannabis trial literature to date is beyond the scope of this review, there are several common themes that limit the implications of these studies. First, many have small sample sizes and are potentially under-powered which increases the likelihood of inappropriately failing to reject the null hypothesis. This issue is not solved by combining multiple studies to increase power, especially in the case of significant heterogeneity between studies [9]. This diversity relates not only to issues with design such as the choice of primary outcome but also variation in terms of disease processes and cannabis formulations studied. For example, a recent meta-analysis examining the effects of cannabis on chronic pain included patients with arthritis, cancer and neuropathic pain, each condition has distinct biologic underpinnings [10]. Furthermore, cannabis formulations ranged from extracts, whole plant and synthetic compounds (very different chemistry) with a variety of administration routes and dosages. Importantly, many of the cannabis formulations studied are not reflective of what is available to patients today including a lack of data on cannabis products that contain high percentages of cannabidiol (CBD). In many reviews, cannabis is simply treated as a monolithic entity with little consideration given to the variety of chemical compounds that have potential physiologic effects. Pooling all of these diseases and drugs together in a single review does little to inform the clinical management of a patient seeking help for symptoms and various illnesses.

## 3. The Chemistry of Cannabis

As cannabis becomes a legitimate medical substance, its chemical consistency is critical for developing cannabis-based pharmaceutical products used in patient care for different medical purposes. Although cannabis strains were originally thought to be either ‘Sativa’ or ‘Indica’ cultivars, the extensive breeding of the cannabis plant has yielded cannabis into a hybrid chemotype of various strains. Licensed producers have used distinct agricultural processing practices to fabricate a variety of cannabis strains with heterogenous chemical compositions and concentrations [11]. This has resulted in numerous available strains and often significant batch-to-batch variation in the marketplace. These variations impact the potency, pharmacokinetics and physiological effects of cannabis-based products. Therefore, it is important to identify and classify medical cannabis strains by their chemistry (i.e., chemovar) rather than their imprecise breeding pattern or name. Given the lack of standardization of the chemical constituents within cannabis products, the clinical outcomes for one defined cannabis-based medicine cannot be generalized to other chemovars. Thus, the ability to demonstrate chemical consistency in cannabis products is essential for the quality production of safe and effective cannabis drug development to ensure reproducible effects.

The major advances in cannabinoid research are attributed to the discovery of the endocannabinoid system (ECS), which is composed of endogenous ligands (primarily anandamide and 2-Arachidonoylglycerol), enzymes and cannabinoid receptors (CBR) that are widely distributed throughout the human body [12,13,14]. The ECS plays vital regulatory roles in multiple physiological processes and behavioral pathways related to inflammatory and immune regulation, synaptic plasticity and neuroprotection, cancer progression, appetite and metabolism, thermogenesis, learning and memory, circadian rhymes, pain regulation, mood and behavior, reproduction, gametogenesis and heart function [15,16,17,18,19,20,21,22,23,24,25]. Recent studies have shown that the ECS may potentially undergo epigenetic modulation through various environmental and lifestyles factors that primarily target genes encoding for CBRs and that endocannabinoids may themselves induce epigenetic alterations [26,27]. Although ECS activity impairment has been implicated in several pathological conditions, it further emphasizes the relevance of the ECS as a potential therapeutic target for disease treatment [28,29].

To date, two CBR subtypes have been identified, CB_1_ receptors (cloned in 1990) and CB_2_ receptor (cloned in 1993), which differ in signaling mechanisms, tissue distribution, agonists/antagonists sensitivity and an amino acid sequence with CB_2_ sharing only 44% of amino acid sequence identity with CB_1_ [30,31,32]. The CBRs are a class of cell membrane receptors that belong to the G-protein coupled receptors (GPCRs) family with their endogenous ligands appearing to be from a family of anandamides [30] More recently, a third CBR known as GPR55 has also been proposed as a possible target for cannabinoid activity but this remains controversial [33]. CB_1_ receptors are the most abundant GPCR, which are primarily expressed throughout the gastrointestinal tract and the central nervous system (CNS) on the axons and axon terminals of neurons predominantly located in the cingulate gyrus, prefrontal cortex, hippocampus, cerebellum and basal ganglia [34]. Activating the CB_1_ receptors in the presynaptic terminal inhibits adenylate cyclase and the cyclic adenosine monophosphate (cAMP) and protein kinase A pathway, resulting in A-type potassium channel activation and subsequently inhibiting various neurotransmitters and neuromodulators [30,35]. CB_2_ receptors are predominately expressed on immune cells and tissues, including the thymus, spleen, tonsils, lymphocytes and macrophages [30,31,36]. More recently, CB_2_ receptors have also been shown to be present in the CNS (i.e., neurons of the hippocampus and microglial cells) in low concentrations [37]. Similar to CB_1_, activating CB_2_ inhibits adenylate cyclase and initiates mitogen-activated protein kinase, leading to increased intracellular calcium levels [30,35]. One of the functions of CB_2_, along with CB_1_, in the immune system is to modulate cytokine release [38]. Activating CB_2_ also induces apoptosis, suppresses autoantibodies and cell proliferation and provides an overall inhibitory effect on inflammatory processes [39].

The majority of the chemical constituents of cannabis are produced by the female *Cannabis sativa* plant, which is classified into hundreds of varieties based on the composition of its fatty compounds known as phytocannabinoids [40,41]. All phytocannabinoids in cannabis are highly lipophilic molecules with low aqueous solubility [40,41]. Phytocannabinoids exert their effects on the ECS through multiple receptors, including CBRs, opioid and serotonin receptors, adrenergic receptors and GPCRs [12,42,43]. To date, approximately 568 compounds have been identified within the cannabis plant, of which roughly 120 are active phytocannabinoids that are synthesized in the secretory cells inside the glandular trichomes [40,41]. These naturally occurring phytocannabinoids share common chemical structural features (i.e., dibenzopyran ring and a hydrophobic alkyl chain) and are distributed among subclasses, including delta-9-tetrahydrocannabinol (Δ*^9^*-THC), Δ*^9^*-tetrahydrocannabinol (Δ*^8^*-THC), CBD, cannabigerol (CBG), Δ*^9^*-tetrahydrocannabivarin (THCV), cannabivarin (CBV), cannabidivarin (CBDV), cannabinol (CBN), cannabichromene (CBC), cannabinodiol (CBND), cannabielsoin (CBE), cannabicyclol (CBL), cannabichromene (CBC), cannabitriol (CBT) and over 30 other miscellaneous types that have undetectable levels in many commercial cannabis chemovars (Figure 1) [44]. Of these phytocannabinoids, the most chemically abundant and well-studied are THC and CBD. Although THC and CBD can be found in the dried cannabis plant, both are naturally produced in their carboxylic acid form, as tetrahydrocannabinol acid (THCA) and cannabidiol acid (CBDA), which are synthesized from a common precursor, cannibigerolic acid (CBGA) (Figure 2) [45,46].

One of the main challenges in addressing cannabis as a therapeutic is that, in contrast to traditional pharmaceuticals, cannabis contains several bioactive compounds and thus behaves as a complex mixture of active pharmaceutical ingredients (APIs). In addition to the extensively studied effects of THC and CBD, minor phytocannabinoids as well as non-cannabinoid phytochemicals such as CBC, CBG and beta-caryophyllene have also exhibited binding to cannabinoid receptors [47,48,49] and it remains to be determined how these activities contribute to or modulate the receptor activity of cannabis as a whole. Further complicating matters is the diversity of cannabis chemovars available to patients. The phytocannabinoid composition of cannabis is generally dominated by the THC and CBD compound families (i.e., THC, CBD and their acidic precursors THCA and CBDA) [50]. However, even among these major phytocannabinoids, significant variation in content is observed across cannabis varieties. In an analysis spanning 277 medical cannabis products from 25 licensed Canadian producers, Mammen et al. noted that THC content varied from 0.14% to more than 25% [51]. Meanwhile, the chemical quantification of 47 medical cannabis extracts and 12 cannabis oils by Yang and colleagues showed that CBD, THC, CBDA and THCA concentrations ranged from 0–51.7%, 0–73.1%, 0–60.3% and 0–76.1% (*w*/*w*), respectively, in the extracts and 0–8.9%, 0–4.6%, 0–31.7% and 0–15.3% (*w*/*v*), respectively, in the oils [52]. Variance in the chemical composition of medical cannabis products inevitably leads to variance in pharmacological responses when these products are consumed by patients [52].

Over the past few years, attempts have been made to deconvolute the chemical complexity of medical cannabis. Recently, Yang et al. pioneered a chemoinformatic approach, whereby the chemical profiles of medical cannabis extracts and cannabis oils were correlated to their corresponding in-vitro cannabinoid receptor activities. This regression analysis generated models which can predict the CB_1_ and CB_2_ activities of cannabis extracts and oils based on their concentrations of 4 major phytocannabinoids—CBD, THC, CBDA and THCA [52]. While such modeling at the in-vitro level has only limited use in a clinical context, it provided evidence that phytocannabinoids may behave differently when administered as part of an extract, compared to when administered as a pure compound. It is thus hoped that in addition to receptor activities, the inclusion of biomarkers and clinical markers such as cytokine levels and von Frey filament measurements as dependent variables could lead to more sophisticated modeling that accurately elucidates the relationship(s) between the chemical makeup of medical cannabis and their corresponding in-vivo clinical outcomes for ailments such as inflammation and chronic pain. Presently, healthcare professionals play an auxiliary role in the administration of medical cannabis, which is largely self-titrated by patients [53,54]. As such, more advanced and clinically relevant models could serve to guide physicians in their medical cannabis prescriptions, by narrowing down potential effective chemovars based on the type and severity of symptoms experienced by patients.

THC is the most prevalent phytocannabinoid in drug chemotypes, which is primarily metabolized in the liver through cytochrome (CYP) 3A4 and CYP2C9 [55]. Chemically, THC has a tricyclic 21-carbon structure without nitrogen and with two chiral centers in trans-configuration [12]. The decarboxylation of THCA to active THC occurs gradually in the dried cannabis plant but can be accelerated upon heating (i.e., smoking, baking or vaporization) [45,46]. This decarboxylation step is essential for obtaining higher potency and binding affinity at CBRs for achieving reliable pharmacological effects when cannabis and its derivatives are used for their therapeutic purposes [11,45,46]. THC, along with CBN and CBND, are known to induce a psychotropic effect [12]. This occurs as THC, being a CB_1_ agonist and a weak CB_2_ partial agonist, activates CBRs through the attenuation of neuronal mechanisms that stimulate receptor downregulation and inhibits the release of neurotransmitters, resulting in a psychotropic high [56]. Various ligand binding studies have indicated that THC has a high affinity to both CB_1_ (Ki = 53 nM) and CB_2_ receptors (Ki = 40 nM) but mainly binds to CB_1_ receptors due to higher receptor affinity [57].

CBD is the second most prevalent phytocannabinoid in drug chemotypes that is metabolized by the cytochrome P450 pathway by several isozymes [55]. It has a bi-cyclic 21-carbon structure with a double bond in the terpene ring [12]. The decarboxylation of CBDA to active CBD also occurs gradually but can be expedited upon heating [11]. CBD is a non-psychotropic phytocannabinoid and at high doses can modulate the intoxicating effects of THC through CB_1_ dependent mechanisms despite having low CB_1_ binding affinity [58]. This observed CB_1_ antagonism may be related to CBD’s ability to behave as a negative allosteric modulator of THC at CB_1_ [59]. CBD is an antagonist of CBRs that only indirectly interacts with CBRs at high concentrations and has been reported to have relatively weak binding affinities for CB_1_ (range, 4.35–10 nM) and CB_2_ (range, 2.4–10 nM) [11]. Unlike THC, studies have proposed that CBD also binds to other non-CBR types of receptors with low efficiency, including serotonin 1A receptor, vanilloid receptor 1 and adenosine A2A receptors [12,42,43].

Cannabis also contains other organic compounds, such as flavonoids and terpenoids, which may contribute to cannabis chemical activity. Flavonoids impart cannabis’ color while terpenoids are responsible for the taste and aroma of cannabis [60]. Both compounds vary substantially across different cannabis chemovars. Over 200 terpenoids have been reported with approximately 50 cannabis terpenes routinely encountered in North America chemovars [60,61]. In the majority of cannabis samples, terpenoids consist of <1% of cannabis with the most common terpenes being myrcene, pinene, terpinolene, limonene, β-caryophyllene and linalool (Figure 3) [60,61]. Although both flavonoids and terpenoids can be bioactive, minimum research has been conducted on their physiological and pharmacological effects in humans. Collectively, when active and inactive cannabis metabolites interact to affect receptor potencies of the active constituents, this is referred to as an “entourage effect.” It is believed that the entourage effect can influence the subjective experience or clinical effects of cannabinoids by non-cannabinoid compounds [60,61]. This suggests that cannabis chemovars may differ in their clinical applications due to their unique combination of cannabinoid and non-cannabinoid components.

From a chemical perspective, exogenous CBR ligands can be categorized into three primary types—phytocannabinoids, endocannabinoids and synthetic cannabinoids (SC) (Figure 4). As mentioned above, phytocannabinoids are the natural plant-derived compounds found in the cannabis plant. Endocannabinoids are naturally occurring cannabinoid molecules that are synthesized in the living organism [62]. The two main and most well-studied endocannabinoids are anandamide and 2-Arachidonoylglycerol, which partially overlap some of the behavioral and molecular characteristics of THC [13,14,63]. Following the isolation and characterization of THC, SC derivatives were also developed. SC, such as dronabinol, nabilone and nabiximols are laboratory fabricated structural THC analogues that have gained widespread attention for their utilization in various disease aliments as they closely mimic the effects of THC on the ECS [62]. Although SC are considered chemical relatives to cannabis, differences do exist among SC and plant-based cannabis. First, THC in natural cannabis is considered a partial agonist to CBRs, whereas SC are full agonists with higher affinity for CBRs [64]. Second, SC are produced as either oral capsules or sprays, which alter their pharmacokinetics and pharmacodynamics [65]. For instance, dronabinol, an oral synthetic THC capsule, has poor solubility and its high first-pass hepatic metabolism accounts for its poor bioavailability with only 10–20% of the administered dose reaching systemic circulation [65] Dronabinol and its active metabolite, 11-hydroxy-delta-9-tetrahydrocannabinol (11-hydroxy-THC), also have a slow absorption rate with peak plasma concentrations being obtained at 2–4 h after consumption that declines steadily over several days [65]. The compound 11-hydroxy-THC can also be more potent than THC and produced in larger quantities, which can heighten SC psychoactive and adverse effects [66,67]. In contrast, inhaled or vaporized cannabis is rapidly absorbed and fast-acting as it bypasses gastrointestinal absorption and hepatic metabolism [66,67]. Third, aside from THC, SC lacks all other active ingredients found in cannabis (e.g., terpenoids) [64]. Given this, SC may be less effective and contribute to an increased psychoactive effect as it does not contain elements to moderate these chemical effects and possibly prevents the entourage effect. SC products also contain additives and chemical preservatives, which may alter their intended effects [64].

## 4. More Research is Needed

The consistent call within the literature to date is for more evidence and further research. An obvious but potentially misguided place to start would be to conduct more randomized controlled trials (RCTs). The reason is that the problem of heterogeneity described above is not restricted to meta-analyses but can influence the results of individual RCTs. RCTs are undoubtedly the gold-standard when it comes determining the efficacy of a potential therapeutic agent. Yet the results can be influenced by altering the population studied and the way the intervention is administered. At a simple level, any intervention or drug can cause harm, benefit or have no effect. The results that are outputted in a trial represent the aggregate collection of these individual responses or the average effect across a population. At the same time, what matters to clinicians are the implications for a particular patient. Therefore, it behooves us to determine what sub-groups of patients respond to what types (i.e., the chemical composition) of cannabis. This is clearly easier said than done. There are numerous patient characteristics that can influence the impact of a therapy including genetic, psychosocial and underlying disease processes. This is matched by various chemovars of cannabis that can be further modified by dose and mode of administration. The number of permutations is daunting and exceeds what can be effectively discerned in a traditional RCT due to cost and time constraints.

## 5. Moving Forward

In order to make an RCT effective and efficient, there must be a plausible hypothesis with sound justification for a potential effect. It is costly and inefficient to simply guess that random forms of cannabis can be used for random conditions and perform a trial. This justification can and often comes from the realm of basic science especially in cases of novel agents. Drugs are tested on various pre-clinical models before being trialed in humans. This is an important step and there is some evidence that specific chemical compounds hold therapeutic promise. But, as noted in the introduction, cannabis is not new and already widely used. There are large cohorts of patients who can provide important data regarding the effectiveness of cannabis. This real-world evidence has the potential to provide invaluable insights into the therapeutic potential of cannabis, however, to harness it there must be granular data collection regarding the type and amount of cannabis used, as well as outcomes. With respect to the amount of cannabis consumed and the constituents within each product, products continue to be labeled with a recreational marketing angle. For example, there is significant focus on the terpenes and flavonoids which conveys specific aromas and tastes respectively on product information labels, with little evidence of a therapeutic benefit from these compounds. The labelling of *Cannabis sativa* oil products with the percentage of CBD/CBDA or THC/THCA without a distinction of the amount of product that is decarboxylated (hence bioactive) is not helpful to the clinician authorizing the product or the consumer. Standardized labelling guidelines must evolve as more knowledge is gained and the medical cannabis industry becomes sophisticated. Removal of the Indica/Sativa distinctions once a product has been chemically extracted seems prudent and moving to understanding the amount in milligrams of the various cannabinoids (i.e., THC, CBD, CBG, CBN, CBDA, THCA etc.) within each product is essential as we move to better understand what drives the clinical efficacy associated with plant based *Cannabis sativa* products. Only with an understanding of the products being consumed by patients and the implementation of clinically and psychometrically validated questionnaires in routine follow-up can we start to better understand the critical chemical constituents driving clinical care. However, observational studies do have down sides. For example, they are subject to numerous potential biases that even the most sophisticated statistical techniques cannot account for including selection bias, unmeasured confounding and misclassification [68]. At the same time, they provide a cost-effective starting point to inform future (and more costly) RCTs. This is a vital first step and an opportunity to rapidly collect data on numerous patients taking cannabis.

## 6. The Present and Next Steps

In the absence of informative randomized controlled trials, how should clinicians counsel their patients on cannabis use? There are guidelines and suggestions that have been previously written [54] but we will provide some overarching principles. First, it is important to stress that cannabis should not considered a panacea for the majority of conditions. Instead, it should be viewed as an adjunct to other therapies which have been established based on years and decades of studies. This involves setting appropriate expectations regarding the possibility of no effect, especially considering the potential financial burden associated with medical cannabis use. Second, until we have better data, providers must guide patients through a process of trial-and-error, with careful attention paid to clinical improvement and adverse effects. It is simply not possible to authorize cannabis use for a patient and then forget about it. Routine follow-up visits over several months will be required while physicians titrate different cannabis products and dosages until a patient gains a stable effect. Third, clinicians must educate themselves regarding the products available in their marketplace. North America has invested a lot of time and resources into creating a legal framework and an industry to support physician authorization of cannabis but very little investment has gone into educational platforms for the physicians and the public. The above guidance as unsatisfying as it may seem is a real-time reflection of the current state of evidence and clinician practice. Through a formalized and granular data collection process within which we know the chemical inputs (i.e., what people are actually consuming from chemical composition standpoint) for patients already using or beginning to use medical cannabis, we can come closer to the ability to provide targeted clinical decision making and design future appropriate randomized controlled trials.

## Figures and Tables

**Figure 1 molecules-25-04042-f001:**
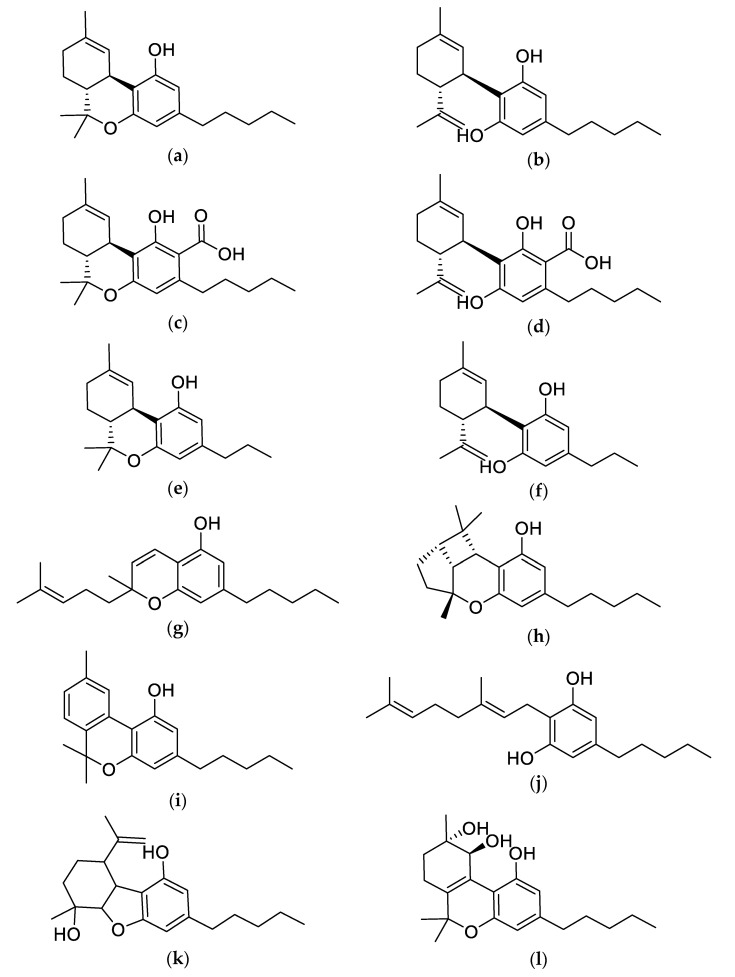
Chemical structures of major phytocannabinoids. (**a**) Δ*^9^*-Tetrahydrocannabinol (Δ*^9^*-THC); (**b**) Cannabidiol (CBD); (**c**) Δ*^9^*-Tetrahydrocannabinolic Acid (Δ*^9^*-THCA); (**d**) Cannabidiolic Acid (CBDA); (**e**) Δ*^9^*-Tetrahydrocannabivarin (THCV); (**f**) Cannabidivarin (CBDV); (**g**) Cannabichromene (CBC); (**h**) Cannabicyclol (CBL); (**i**) Cannabinol (CBN); (**j**) Cannabigerol (CBG); (**k**) Cannabielsoin (CBE); (**l**) Cannabitriol (CBT).

**Figure 2 molecules-25-04042-f002:**
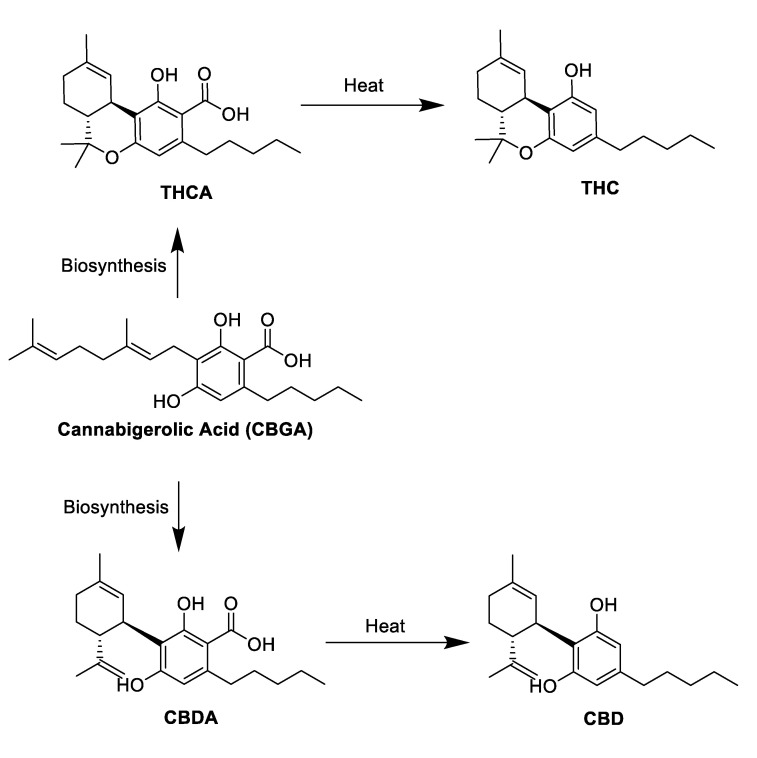
Biosynthesis of the acidic phytocannabinoids Δ*^9^*-THCA and CBDA and their decarboxylation into their active forms Δ*^9^*-THC and CBD.

**Figure 3 molecules-25-04042-f003:**
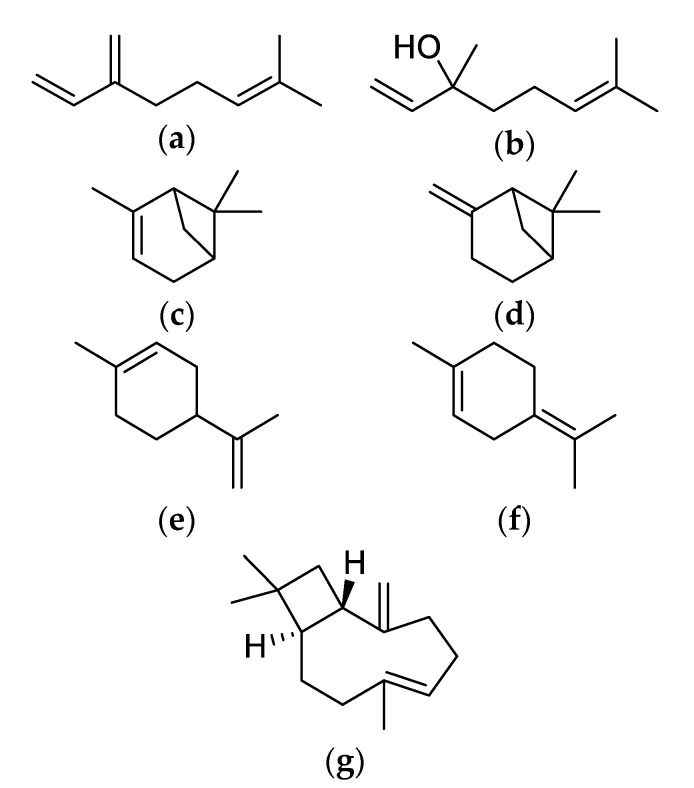
Chemical structures of major non-cannabinoid phytochemicals. (**a**) Myrcene; (**b**) Linalool; (**c**) α-Pinene; (**d**) β-Pinene; (**e**) Limonene; (**f**) Terpinolene; (**g**) β-Caryophyllene.

**Figure 4 molecules-25-04042-f004:**
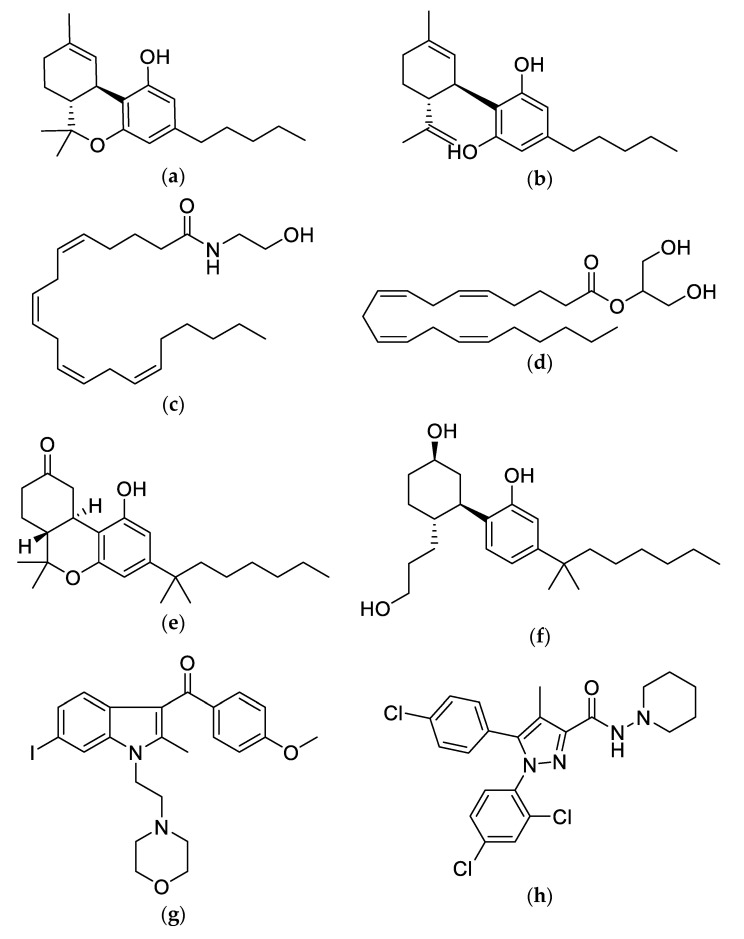
Chemical structures of the phytocannabinoids (**a**) Δ*^9^*-THC and (**b**) CBD, the endocannabinoids (**c**) Anandamide (AEA) and (**d**) 2-Arachidonoylglycerol (2-AG) and the synthetic cannabinoids (**e**) Nabilone, (**f**) CP 55,940, (**g**) AM-630 and (**h**) SR141716. Dronabinol is a synthetically produced form of Δ*^9^*-THC.

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
