# Peer review of "Understanding the Medical Chemistry of the Cannabis Plant is Critical to Guiding Real World Clinical Evidence"

_molecules, 2020, doi:10.3390/molecules25184042_

Round 1
Reviewer 1 Report
There are numerous reviews in the cannabis field, however unfortunately I didn't find any novelty in this review. There was no original point of view or any novel perspective which set it apart from the other reviews published in the field .
I normally am not so blunt and very supportive of this subject, however sadly I feel that the authors missed the major point of summarizing the main knowledge and obstacles of the complex chemistry of the cannabis plant. The authors should summarize the recent developments and approaches that deal with this subject and to discuss how it can revolutionize the entire field giving rise to more accurate understandings and enable important clinical advances.
I found the review to be unfocused and jump between subjects (chemistry, endocannabinoid system, the world situation…) without proper connections and without enough in depth analysis and the required discussion of each subject.
Author Response
We would like to thank the reviewer for their candidness in their feedback and highlighting some of the deficiencies in our article. We have now incorporated an in-depth discussion on the challenges of navigating the complex chemistry of medical cannabis, while highlighting efforts to deconvolute this complexity through chemoinformatic methods.
Reviewer 2 Report
The perspective “Understanding the Medical Chemistry of the Cannabis Plant is Critical to Guiding Real World Clinical Evidence” is brief, well-edited, and makes an important argument. A bit of only minor feedback and suggestions follow.
Minor comments:
L5: Please supply affiliations for the 2nd & 3rd authors
L18: decades<comma>
L24: This is a run-on sentence. Consider “can we” and refining
L44-46: Consider: “In this review, we argue that the current evidence-base lacks relevance and new data generated from prospective observational studies, i.e. real-world evidence, is a crucial first step towards informing future trials.”
L50: double-check journal policies on use of 5 versus five
L52: performance<comma>
L95: The description of receptors is variable in the literature. Please double-check the International Union of Basic & Pharmacology and follow their conventions thoughout (suspect its subscript 1/2 and no “R”).
Pertwee RG, Howlett AC, Abood ME, Alexander SP, Di Marzo V, Elphick MR, Greasley PJ, Hansen HS, Kunos G, Mackie K, Mechoulam R, Ross RA. International Union of Basic and Clinical Pharmacology. LXXIX. Cannabinoid receptors and their ligands: beyond CB₁ and CB₂. Pharmacol Rev. 2010 Dec; 62(4):588-631.
L190: A citation for this “First, THC in natural cannabis is considered a partial agonist to CBRs, whereas SC are full agonists with higher affinity for CBRs. “ is needed.
L208: delete :
L226: efficient<comma>
L251: a bit more detail of example potential biases would be beneficial
L271: citations/links to these modules?
L310: delete San Diego California. If the original citation uses subscripts, this should be retained here (L321, L323 too).
L366: Number of authors pre et al. should be consistent. Double-check journal policy for article title (lower-case?) too.
L368: not sure if this was an attempt at humor?
A graphical abstract was not available in the reviewer copy.
Author Response
L5: Please supply affiliations for the 2nd & 3rd authors
Response: The affiliations have been updated.
L18: decades<comma>
Response: This has been changed.
L24: This is a run-on sentence. Consider “can we” and refining
Response: We have refined this sentence and broken it up into two separate sentences.
L44-46: Consider: “In this review, we argue that the current evidence-base lacks relevance and new data generated from prospective observational studies, i.e. real-world evidence, is a crucial first step towards informing future trials.”
Response: Thank you for this suggestion, we have changed this sentence accordingly.
L50: double-check journal policies on use of 5 versus five
Response: We have changed it to “five”
L52: performance<comma>
Response: The comma has been added.
L95: The description of receptors is variable in the literature. Please double-check the International Union of Basic & Pharmacology and follow their conventions thoughout (suspect its subscript 1/2 and no “R”).
Pertwee RG, Howlett AC, Abood ME, Alexander SP, Di Marzo V, Elphick MR, Greasley PJ, Hansen HS, Kunos G, Mackie K, Mechoulam R, Ross RA. International Union of Basic and Clinical Pharmacology. LXXIX. Cannabinoid receptors and their ligands: beyond CB₁ and CB₂. Pharmacol Rev. 2010 Dec; 62(4):588-631.
Response: We have used the subscripts ½ throughout the manuscript and removed R.
L190: A citation for this “First, THC in natural cannabis is considered a partial agonist to CBRs, whereas SC are full agonists with higher affinity for CBRs. “ is needed.
Response: Thank you for bringing this oversight to our attention. The citation is now added.
L208: delete :
Response: The colon has been deleted.
L226: efficient<comma>
Response: We have added the comma.
L251: a bit more detail of example potential biases would be beneficial
Response: Thank you for this suggestion. We have included some examples of theses biases and also inserted a citation should the reader want further details that are beyond the scope of this manuscript.
L271: citations/links to these modules?
Response: The online modules that have been created are not currently available to the public yet and thus we are not able to provide a citation. We have removed this sentence to avoid confusion for the reader.
L310: delete San Diego California. If the original citation uses subscripts, this should be retained here (L321, L323 too).
Response: This has been changed according to the reviewer’s comment.
L366: Number of authors pre et al. should be consistent. Double-check journal policy for article title (lower-case?) too.
Response: We have reformatted the references to be consistent with journal policy.
L368: not sure if this was an attempt at humor?
Response: Thank you for pointing this out. This seems to have been added by the submission system. We have removed this.
A graphical abstract was not available in the reviewer copy.
Response: We have not created a graphical abstract, but happy to do so if the editor requests one.
Reviewer 3 Report
The submitted review article provides a critical overview on the use of cannabis for therapeutic purposes. This upcoming issue is really interesting and deserves attention. I only suggest few improvements.
· Apart from anamdamide, also 2-Arachidonoylglycerol (2-ag) is considered one of the main endocannabinoids.
· The description of the functions of ECS is quite limited (i.e CB2 described in relationship to immune functions and CB1 in relationship to brain, spinal cord and gastrointestinal tract). Please include a more broad description including functions (and references) related to reproduction, behaviour, cancer, food intake, cardiolvascular system and so on. Furthermore, take into account a recently published special issue from MDPI entitled " Endocannabinoid System in Health and Disease: Current Situation and Future Perspectives".
· Following the description of ECS functions, please include its activity as "epigenetic" target and the possibility that endocannabinoids may themselves induce epigenetic changes, an issue not secondary in therapy.
- 13 correct Δ9_THC
- among the main endocannabinoids cite 2-Arachidonoylglycerol (2-ag)
- cannabis sativa is the name of a specie thus require Italic font (Cannabis sativa)
- Some abbreviations have not been defined at the first appearance in the main text (examples lane 73).
Author Response
The submitted review article provides a critical overview on the use of cannabis for therapeutic purposes. This upcoming issue is really interesting and deserves attention. I only suggest few improvements.
- Apart from anamdamide, also 2-Arachidonoylglycerol (2-ag) is considered one of the main endocannabinoids.
Response: Thank you. We have now included this into the manuscript.
The description of the functions of ECS is quite limited (i.e CB2 described in relationship to immune functions and CB1 in relationship to brain, spinal cord and gastrointestinal tract). Please include a more broad description including functions (and references) related to reproduction, behaviour, cancer, food intake, cardiolvascular system and so on. Furthermore, take into account a recently published special issue from MDPI entitled " Endocannabinoid System in Health and Disease: Current Situation and Future Perspectives".
Response: Thank you for this suggestion, we have now incorporated the requested descriptions and information in the manuscript.
Following the description of ECS functions, please include its activity as "epigenetic" target and the possibility that endocannabinoids may themselves induce epigenetic changes, an issue not secondary in therapy.
Response: Thank you, we have now added this to the manuscript.- 13 correct Δ9_THC
Response: Thank you for this. We have now corrected this in the manuscript
- among the main endocannabinoids cite 2-Arachidonoylglycerol (2-ag)
Response: Thank you. We have now included this into the manuscript.
- cannabis sativa is the name of a specie thus require Italic font (Cannabis sativa)
Response: Thank you for addressing this, we have now corrected for this in the manuscript.
- Some abbreviations have not been defined at the first appearance in the main text (examples lane 73).
Response: Thank you, we have now corrected this in the manuscript.
Reviewer 4 Report
The brief communication entitled “Understanding the Medical Chemistry of the Cannabis Plant is Critical to Guiding Real World Clinical Evidence” highlights some important points about research in the cannabinoid field and identifies critical data gaps. The manuscript is timely and well-written and I think could be of interest to many in the field, especially clinicians.
Because of its expected significance, perhaps the authors could consider pulling out important concepts to be highlighted; for instance, could the authors could define the term “chemovar” in a box? This is a critical term that is not widely known, even in the field.
Also perhaps a box at the end with some critical bullet points from sections 4 and 5 about RCTs, identification of chemicals being tested and advice for clinicians.
Author Response
Response: We thank the reviewer for these excellent suggestions. From reading the instructions to authors on the journal’s website, it does not appear that we are able to insert highlight tables into this manuscript. However, we are happy to do so if the editor allows us to as we agree that it would help highlight key points of the article.
This manuscript is a resubmission of an earlier submission. The following is a list of the peer review reports and author responses from that submission.